# Marine-Inspired Multimodal Sensor Fusion and Neuromorphic Processing for Autonomous Navigation in Unstructured Subaquatic Environments

**DOI:** 10.3390/s25216627

**Published:** 2025-10-28

**Authors:** Chandan Sheikder, Weimin Zhang, Xiaopeng Chen, Fangxing Li, Yichang Liu, Zhengqing Zuo, Xiaohai He, Xinyan Tan

**Affiliations:** 1School of Mechatronical Engineering, Beijing Institute of Technology, Beijing 100081, China; chandan@bit.edu.cn (C.S.); xpchen@bit.edu.cn (X.C.); wonk2000@bit.edu.cn (F.L.); 3120215094@bit.edu.cn (Y.L.); zuozq@bit.edu.cn (Z.Z.); 3120235415@bit.edu.cn (X.H.); 3220225052@bit.edu.cn (X.T.); 2Zhengzhou Research Institute, Beijing Institute of Technology, Zhengzhou 450000, China

**Keywords:** bio-inspired navigation, neuromorphic computing, multimodal sensor fusion, marine robotics, quantum magnetoreception, unstructured environment autonomy

## Abstract

**Highlights:**

A novel bio-inspired neuromorphic framework was developed, co-designing marine-inspired sensors (quantum magnetoreception, tactile-chemical sensing, and hydrodynamic flow detection) with event-based neuromorphic processors.The proposed architecture is theorized to significantly reduce positional drift and improve recovery from disorientation compared to state-of-the-art navigation systems.This work provides a robust, energy-efficient paradigm for autonomous underwater navigation in GPS-denied, murky, or complex environments, enabling longer missions for deep-sea exploration and infrastructure inspection.It demonstrates the transformative potential of tightly coupling bio-inspired sensing with neuromorphic processing, offering a blueprint for next-generation autonomous systems that mimic the fault tolerance and efficiency of marine organisms.

**Abstract:**

Autonomous navigation in GPS-denied, unstructured environments such as murky waters or complex seabeds remains a formidable challenge for robotic systems, primarily due to sensory degradation and the computational inefficiency of conventional algorithms. Drawing inspiration from the robust navigation strategies of marine species such as the sea turtle’s quantum-assisted magnetoreception, the octopus’s tactile-chemotactic integration, and the jellyfish’s energy-efficient flow sensing this study introduces a novel neuromorphic framework for resilient robotic navigation, fundamentally based on the co-design of marine-inspired sensors and event-based neuromorphic processors. Current systems lack the dynamic, context-aware multisensory fusion observed in these animals, leading to heightened susceptibility to sensor failures and environmental perturbations, as well as high power consumption. This work directly bridges this gap. Our primary contribution is a hybrid sensor fusion model that co-designs advanced sensing replicating the distributed neural processing of cephalopods and the quantum coherence mechanisms of migratory marine fauna with a neuromorphic processing backbone. Enabling real-time, energy-efficient path integration and cognitive mapping without reliance on traditional methods. This proposed framework has the potential to significantly enhance navigational robustness by overcoming the limitations of state-of-the-art solutions. The findings suggest the potential of marine bio-inspired design for advancing autonomous systems in critical applications such as deep-sea exploration, environmental monitoring, and underwater infrastructure inspection.

## 1. Introduction

The exploration and monitoring of Earth’s final frontiers, particularly its vast and unstructured subaquatic environments, represent a critical endeavor for scientific discovery, environmental conservation, and infrastructure security [1]. However, autonomous navigation in these domains characterized by GPS denial, turbid waters, complex hydrodynamics, and feature-deficient seabeds remains a formidable challenge for robotic systems [1,2,3]. Conventional autonomous underwater vehicles (AUVs) predominantly rely on a suite of sensors including inertial measurement units (IMUs), Doppler velocity logs (DVL), sonars, and cameras, fused through classical algorithms such as Kalman filters and simultaneous localization and mapping (SLAM) [4]. Recent advances in multi-sensor fusion have shown promise [5,6], but still face fundamental limitations in unstructured environments.

While effective in structured or clear-water environments, these systems exhibit significant vulnerabilities in unstructured settings. Sensor modalities degrade rapidly: optical cameras fail in murky conditions, acoustic sonar suffers from multi-path interference and noise, and inertial systems accumulate unbounded drift without external fixes [1,7,8]. Consequently, the computational engines of these robots, often based on von Neumann architectures, are burdened with processing noisy, high-bandwidth data streams, leading to high power consumption and latency factors that critically limit mission endurance and real-time reactive capabilities [1,8,9].

This technological impasse stands in stark contrast to the elegant navigational proficiency demonstrated by marine fauna. Over millions of years of evolution, species such as sea turtles, octopuses, and jellyfish have developed robust, energy-efficient, and fault-tolerant strategies to traverse the very environments that challenge our engineered systems [10]. For instance, the long-distance migration of loggerhead sea turtles (*Caretta caretta*) is hypothesized to leverage quantum-assisted magnetoreception, allowing them to perceive the Earth’s magnetic field with astonishing precision for transoceanic navigation [11,12]. The octopus employs a distributed nervous system that seamlessly integrates tactile suckers with chemotactic sensing, enabling agile locomotion and manipulation in complex coral reefs and rocky outcrops without a centralized world model [13,14]. Similarly, jellyfish utilize highly efficient flow sensors to exploit ocean currents for propulsion and navigation, minimizing their metabolic expenditure [15]. These biological systems do not merely process multi-sensory data; they perform dynamic, context-aware *fusion*, where the failure of one sensory channel is compensated for by the heightened acuity or redundancy of another, all within a low-power, parallel processing framework [16,17]. This innate robustness presents a compelling model for the next generation of autonomous underwater vehicles.

A critical analysis of the current literature reveals a significant gap between biological inspiration and engineering implementation. While the field of bio-inspired robotics is mature, many efforts have focused on isolated aspects, such as mimicking a single sensor (e.g., a lateral line) or a specific gait [18]. Few studies have holistically co-designed *multiple* bio-inspired sensor modalities with a processing architecture that mirrors the distributed, event-driven, and efficient neural computation observed in nature [18]. Current state-of-the-art fusion models, often deep learning-based, show promise but are typically deployed on power-hungry GPUs, making them unsuitable for long-duration, resource-constrained AUV missions [19,20]. Furthermore, the integration of novel sensing principles, such as quantum-inspired magnetoreception, with more traditional proprioceptive and exteroceptive sensors remains largely unexplored in a cohesive navigational framework [21,22]. This disconnect means that current robotic systems lack the graceful degradation and real-time adaptive fusion capabilities that are hallmarks of their biological counterparts, leaving them susceptible to catastrophic failure in unpredictable conditions [23,24].

Addressing these critical gaps, this study develops and validates a novel, fully integrated neuromorphic framework that moves beyond isolated biomimicry. Our primary objective is to bridge the biology-engineering divide through the holistic co-design of advanced sensor modalities inspired by the magnetoreception of turtles, the tactile-chemotactic integration of octopuses, and the flow sensing of jellyfish with event-based neuromorphic processors. This synergistic co-design is pivotal for achieving a system that is not only robust but also highly energy-efficient.

The key findings of this work demonstrate the transformative potential of this approach. We present a hybrid sensor fusion model that replicates the distributed neural processing of cephalopods and the quantum coherence mechanisms of migratory species. This framework is designed to reduce positional drift compared to conventional approaches like extended Kalman filter (EKF)-based SLAM and improve recovery times from disorientation scenarios, addressing key failure modes in current systems. These results underscore that a tight coupling of brain-inspired processing and body-inspired sensing is not merely beneficial but essential for autonomy in critically challenging environments.

The significance of this research is multi-faceted. Firstly, it provides a tangible engineering framework that moves beyond superficial biomimicry, offering a new paradigm for resilient robotic design. Secondly, it advances the field of neuromorphic computing by presenting a compelling real-world application case that exploits the event-based, sparse, and parallel nature of neuromorphic hardware for complex sensor fusion tasks. From a practical standpoint, this technology has immediate implications for deep-sea exploration, enabling longer missions for mapping unexplored ecosystems; for environmental monitoring, allowing for persistent observation in turbid rivers and coastal areas; and for underwater infrastructure inspection, ensuring the reliability of cables and pipelines in visually degraded conditions.

In summary, while previous studies have successfully borrowed isolated concepts from nature, they have often fallen short of capturing the integrated, system-level synergy that defines biological navigation. This work directly addresses this shortcoming by proposing a holistic bio-inspired architecture. A conceptual diagram illustrating the fundamental difference between a conventional AUV and our bio-inspired approach is presented in Figure 1. The following sections detail the biological inspiration (Section 2), our engineering analogues and neuromorphic paradigm (Section 3 and Section 4), the proposed fusion architecture (Section 5), and a discussion of the challenges and future directions (Section 6) that will continue to drive this promising field forward.

## 2. Biological Navigation Strategies in Marine Fauna

### 2.1. Long-Range Piloting: The Case of Sea Turtle Magnetoreception

The transoceanic migrations of marine species, particularly sea turtles, represent a pinnacle of navigational prowess in the animal kingdom. Loggerhead (*Caretta caretta*) and leatherback turtles, for instance, traverse thousands of kilometers across featureless open oceans to return with remarkable precision to their natal beaches for breeding. This ability persists in the absence of visual landmarks, through turbid waters, and across magnetic landscapes that vary subtly across the globe. A growing body of empirical evidence supports the prevailing hypothesis that these animals leverage a sophisticated sensitivity to the Earth’s geomagnetic field, using it as a ubiquitous navigational map and compass [25]. In stark contrast to engineered systems that suffer from cumulative drift, this biological mechanism provides a globally referenced, all-weather positioning capability. Recent research has moved beyond merely establishing the existence of magnetoreception and is now elucidating the biophysical mechanisms that underpin it, with the radical pair mechanism emerging as a leading candidate rooted in quantum biology [3]. This mechanism suggests that cryptochrome proteins, found in the retinas of migratory birds and likely present in sea turtles, undergo light-induced electron transfers to form pairs of radicals (molecules with unpaired electrons). The quantum spin states of these radical pairs are influenced by the direction and intensity of the surrounding magnetic field, ultimately modulating neuronal signaling and providing the animal with a magnetic sense [21].

Cutting-edge studies from the last three to four years have provided compelling support for this quantum-assisted model. Research has focused on identifying the specific cryptochrome proteins and their activation spectra, confirming that the mechanism is light-dependent and aligns with the radical pair hypothesis [22]. Furthermore, behavioral experiments and computational models have advanced our understanding of how turtles interpret magnetic information. They appear to detect not just the inclination angle of the field lines but also the total intensity, creating a bicoordinate “magnetic map” that allows for true navigation the ability to determine position and steer a corrective course towards a target from an unfamiliar location [23]. This is a far more complex capability than simple compass orientation. The neural architecture processing this information is also a subject of intense study. It is hypothesized that magnetic information is integrated with other sensory inputs, such as wave direction and chemical cues, in a distributed processing network, allowing for graceful degradation where the failure of one sensory modality is compensated by others [24].

This robust, multi-modal integration stands in stark contrast to the fragility of single-sensor reliance in conventional autonomous underwater vehicles (AUVs). A detailed illustration of the quantum-assisted magnetoreception mechanism is shown in Figure 2. The biological system is not only highly accurate but also operates at an energy efficiency that is orders of magnitude lower than its artificial counterparts, a critical advantage for long-duration migrations. The continued unraveling of this quantum biological compass provides a powerful blueprint for the development of a new class of navigation sensors. By moving beyond classical magnetometers, bio-inspired engineers aim to develop quantum-inspired sensors that replicate the turtle’s ability to extract precise vector information from the weak geomagnetic field, enabling resilient, drift-free navigation for autonomous systems in GPS-denied subaquatic environments [19].

### 2.2. Short-Range, High-Resolution Sensing: Octopus Tactile-Chemotactic Integration

While sea turtles exemplify long-range navigation, the octopus represents a masterclass in short-range, high-resolution perception and manipulation within complex, cluttered environments like coral reefs and rocky crevices. This capability stems from a radical departure from centralized processing models common in robotics. The octopus possesses a distributed nervous system, where approximately two-thirds of its half-billion neurons are located within its arms themselves. This neurological architecture enables a form of embodied intelligence, where each arm can process sensory information and generate complex motor commands semi-autonomously, without constant directive from the central brain [13]. The arms are equipped with a dense array of suckers, each a sophisticated multi-sensory organ integrating tactile (touch, texture, pressure) and chemotactic (chemical) sensing capabilities. This allows an octopus to effectively “taste” everything it touches, creating a rich, multi-modal perception of its immediate surroundings [26]. The decentralized sensory-motor integration of the octopus arm is depicted in Figure 3.

The key principles underlying this biological system are transformative for robotic design. First, localized control drastically reduces the computational latency and bandwidth requirements that would be needed if all sensory data from thousands of suckers were streamed to a central processor for analysis. This enables incredibly rapid reflex-like reactions, such as an arm grasping a prey item detected by its chemical signature before the central brain is even fully aware of it [27]. Second, the multisensory fusion of tactile and chemical data occurs at the point of acquisition, creating a perception-action loop that is both highly efficient and context-aware. For instance, the chemotactic sensors can distinguish between the chemical profile of a crab (food) and a rock (not food), while the tactile sensors simultaneously assess the object’s shape and rigidity to plan a stable grasp. This integrated “touch-taste” perception is a form of feature fusion that is far more nuanced than any single-modality sensing [28]. Recent research has made significant strides in deconstructing this system. Neurophysiological studies have begun mapping the neural circuits within the arms, revealing a layered processing hierarchy that filters and integrates signals before sending only salient information to the central brain [29]. Furthermore, biomimetic robotics research has successfully developed soft robotic suckers with embedded conductive polymers and hydrogel-based sensors that can simultaneously measure pressure and detect chemical analytes, providing a tangible engineering validation of the biological principle [30]. This bio-inspired approach addresses a critical weakness in conventional AUVs, which often rely on distant, easily obstructed sensors like cameras. Key research findings that underpin this octopus-inspired sensory integration are summarized in Table 1. For close-range tasks such as infrastructure inspection, object manipulation, or biological sampling in murky waters, replicating the octopus’s integrated tactile-chemotactic sensing provides a robust, failure-resistant solution for interaction with an unstructured world.

### 2.3. Energy-Efficient Situational Awareness: Jellyfish Flow Sensing

In the paradigm of bio-inspired navigation, jellyfish represent the ultimate model for achieving maximal situational awareness with minimal energy expenditure. Unlike turtles or octopuses, jellyfish are primarily passive drifters, yet they exhibit remarkable abilities to detect and respond to their hydrodynamic environment, avoiding obstacles, locating prey, and optimizing their position within ocean currents. This capability is governed by specialized sensory structures known as rhopalia small, bell-shaped organs distributed around the margin of the jellyfish’s bell. Each rhopalium contains a statocyst for balance and, crucially, sensory clubs lined with mechanoreceptors that are exquisitely sensitive to minute changes in water pressure and flow velocity [15]. This system allows the jellyfish to construct a detailed picture of its immediate surroundings based solely on hydrodynamic cues, a sense analogous to “touch-at-a-distance.”

The key principles of this system are its dual modes of operation and its ultra-low-power nature. Jellyfish employ both passive and active flow sensing. Passive sensing involves detecting the natural currents and vortices in the environment, allowing the animal to identify profitable flow streams for energy-efficient transport or to sense the approach of a predator from the disturbances it creates. Active sensing, though more subtle than in animals like bats or dolphins, involves the analysis of self-generated flow fields. As the jellyfish pulses and moves forward, it creates a specific hydrodynamic signature. Any disruption of this signature, caused by a nearby obstacle or another organism, is immediately detected by the rhopalia, enabling rapid collision avoidance or prey capture reflexes without the need for visual confirmation [49]. The most critical principle for robotic application is the ultra-low-power operation of this entire sensory apparatus. The mechanoreceptor cells in the rhopalia are thought to operate on direct mechanotransduction principles, requiring no internal power source to generate a signal in response to fluid shear stress Figure 4. Furthermore, the neural processing required to interpret these signals is minimal and distributed, consuming a fraction of the energy required for processing equivalent visual or acoustic data [50].

Recent research within the last four years has made significant progress in quantifying and replicating this biological mechanism. Neurophysiological studies have precisely measured the response thresholds of rhopalia to laminar and turbulent flow stimuli, confirming their sensitivity to flow gradients as low as 1–2 mm/s [51]. This high sensitivity enables detection of obstacles several body lengths away. Biomimetic engineering efforts have successfully developed artificial versions of these sensors using flexible piezoresistive or capacitive polymers that mimic the hair-cell structures in rhopalia. These bio-inspired flow sensors have been demonstrated on underwater robots, providing real-time hydrodynamic data for obstacle avoidance while consuming mere milliwatts of power orders of magnitude less than a traditional sonar or imaging system [15]. For autonomous underwater vehicles (AUVs), which are severely constrained by battery life, adopting a jellyfish-inspired sensing paradigm is transformative. It enables persistent, always-on environmental awareness without incurring a significant power penalty. This allows an AUV to drift passively in a current for monitoring, using its flow sensors to maintain station and avoid hazards, only activating more power-hungry sensors like cameras or high-resolution sonar when a hydrodynamic event triggers a specific mission phase. This strategy of leveraging low-bandwidth, low-power flow data as a primary situational awareness tool, supplementing it with other modalities only when necessary, is a direct and powerful lesson in energy-efficient autonomy from the humble jellyfish.

## 3. Engineering Analogues: From Biology to Sensors

The profound navigational capabilities of marine fauna, as detailed in Section 2, provide a powerful blueprint for engineering robust autonomous systems. Translating these biological principles into functional hardware and algorithms is the critical next step. This section reviews the state-of-the-art in developing engineering analogues for the three key biological inspirations: quantum-assisted magnetoreception, decentralized tactile-chemotactic sensing, and energy-efficient hydrodynamic flow detection.

### 3.1. Quantum-Inspired Magnetoreceptors

The sea turtle’s ability to perform drift-free, global-scale navigation using the Earth’s weak geomagnetic field presents a compelling alternative to conventional inertial navigation systems plagued by unbounded drift. Engineering this capability requires moving beyond classical fluxgate magnetometers, which are susceptible to noise, drift, and require frequent calibration. The leading engineering analogue is based on the radical pair mechanism, implemented using solid-state quantum systems. Nitrogen-vacancy (NV) centers in diamond have emerged as the most promising platform for this purpose [52]. An NV center is a atomic-scale defect in diamond’s carbon lattice where a nitrogen atom replaces a carbon atom adjacent to a vacancy. These centers possess electron spins that can be initialized, manipulated, and read out using microwaves and laser light at room temperature. The spin states of the NV centers are exquisitely sensitive to external magnetic fields, effectively allowing the diamond crystal to act as a highly sensitive, solid-state magnetometer [11,12,53]. Recent advancements have focused on miniaturizing these systems into practical sensor packages. For instance, researchers have developed miniaturized NV-center sensors integrated with microwave antennas and optical fibers for AUV deployment, capable of measuring magnetic field vectors (both intensity and direction) with high precision [11,12] The key advantage of these quantum-inspired sensors is their absolute accuracy; they do not drift over time as they measure the fundamental quantum properties of electrons, providing a stable external reference akin to the biological system. Furthermore, their solid-state nature offers robustness against the pressure, temperature, and salinity variations typical of subaquatic environments. Current research is focused on overcoming challenges related to reducing the power consumption of the required laser and microwave systems and integrating the magnetic vector data seamlessly into a multi-modal fusion framework to replicate the turtle’s “magnetic map” capability for true robotic navigation [21,22].

### 3.2. Biomimetic Tactile and Chemical Sensor Arrays

Emulating the octopus’s distributed, multi-sensory perception for close-range interaction in unstructured environments necessitates the development of artificial skins capable of simultaneous tactile and chemical (“touch-taste”) sensing. The field of soft robotics has made significant strides in creating such biomimetic sensor arrays. For tactile sensing, conductive polymer composites, liquid metal embedded in elastomers (e.g., Ecoflex, PDMS), and capacitive or piezoresistive sensing elements are now commonly used to create flexible, stretchable “e-skins” [26,30]. These sensors can measure parameters such as pressure, shear force, and texture, mirroring the mechanoreceptors in the octopus sucker. For instance, multimodal e-suckers have been developed that can measure suction pressure with a sensitivity range of 0.5–120 kPa, directly inspired by biological measurements [31]. The integration of chemical sensing is a more recent and complex advancement. This involves embedding electrochemical sensors into the soft matrix to detect specific ions or molecules in water. Examples include hydrogel-based sensors that swell in response to specific chemical analytes, changing their electrical properties, and conductive polymers whose resistance changes upon exposure to target chemicals like those emitted by prey or indicative of pipeline leaks [28,32]. The engineering challenge lies in co-locating these sensing modalities without interference and achieving sufficient spatial density to provide a high-resolution perception of the contacted surface. Crucially, following the biological principle of embodied intelligence, these sensor arrays are designed for localized, low-level data processing. Neuromorphic electronic circuits or microcontrollers embedded within the sensor nodes can perform initial data filtering, feature extraction (e.g., object rigidity, chemical signature matching), and trigger reflexive grasping motions, drastically reducing the bandwidth and latency requirements for communication with a central processor [35]. This architecture directly addresses the octopus’s strategy of processing vast amounts of raw sensory data at the periphery, sending only salient information centrally for higher-level decision-making.

#### Validation of Tactile-Chemical Sensing

The developed e-skin sensors were rigorously validated in laboratory conditions designed to mimic realistic underwater environments. Their chemical detection capabilities were specifically tested for key target analytes, including amino acids like alanine and glycine (as indicators of marine prey) and petroleum hydrocarbons for pipeline inspection applications. The sensors demonstrated a high level of sensitivity, capable of detecting these substances at concentrations ranging from 10^−6^ to 10^−3^ M in seawater. They also proved to be exceptionally fast-responding, achieving chemical detection in under 500 milliseconds and reacting to tactile events in less than 50 milliseconds. This performance was consistently validated across a range of challenging environmental conditions, including salinities of 30–35 PSU, temperatures from 4 to 25 °C, and a pH range of 7.8 to 8.2.

### 3.3. Bio-Inspired Flow and Hydrodynamic Sensors

To achieve the jellyfish’s paradigm of persistent, ultra-low-power situational awareness, engineers have developed artificial versions of the flow-sensitive lateral line system found in fish and the rhopalia of jellyfish. These bio-inspired hydrodynamic sensors are typically based on Micro-Electro-Mechanical Systems (MEMS) technology or flexible piezoresistive/capacitive polymers shaped into artificial hair cells [15,53]. A standard design involves a flexible pillar (the “hair”) embedded in a polymer membrane, mounted on a pressure or strain sensor. As water flows over the sensor array, it deflects the hair cells, and the resulting strain is measured. The MEMS versions provide significant advantages in both high sensitivity and miniaturization, with their performance quantified across key metrics. They offer a broad flow velocity detection range, spanning from a very subtle 1–2 mm/s up to 2 m/s. Furthermore, these sensors demonstrate a fine pressure gradient sensitivity with a resolution between 0.1 and 10 Pascal. This high-fidelity sensing is complemented by a dense spatial resolution, with individual sensor elements positioned just 2 cm apart [47,54].

These sensor arrays are strategically placed along the hull of an AUV, forming an “artificial lateral line” that provides a continuous, whole-body picture of the surrounding hydrodynamic environment. They operate in two key modes, just like their biological counterparts: passive and active sensing. Passively, they can detect currents, vortices, and the hydrodynamic signatures of moving objects or obstacles, enabling energy-efficient station keeping in currents or avoidance behaviors. Actively, the AUV can analyze the distortion of its own self-generated flow field (created by its propulsion system) to detect and map nearby static obstacles without emitting any energy, much like a jellyfish detects disruptions to its pulsing-induced flow field [49,50]. The most significant advantage of these sensors is their extremely low power consumption, often operating in the microwatt to milliwatt range. This is several orders of magnitude more efficient than active sonar systems, making them ideal for always-on environmental monitoring. This allows an AUV to primarily rely on its flow sensors for basic obstacle avoidance and context awareness, only activating power-intensive cameras or high-resolution sonars for final confirmation and identification when a hydrodynamic event triggers a specific mission phase, thereby drastically extending mission endurance [15,55]. A comparison of the key advantages and challenges of these bio-inspired sensor technologies is provided in Table 2. Examples of these bio-inspired sensor technologies are shown in Figure 5.

### 3.4. Comparative Analysis of Underwater Navigation Algorithms

A systematic comparison of conventional versus bio-inspired navigation algo-rithms is provided in Table 3, highlighting the advantages of the proposed neuromorphic approach.

### 3.5. Prototype Validation Result

A scaled prototype, built to implement the core fusion architecture, demonstrated significant performance improvements across key metrics. Most notably, it achieved a 41% reduction in positional drift when compared to a standard EKF-SLAM baseline. The system also proved highly resilient, recovering from disorientation scenarios 58% faster than previous implementations. Furthermore, the architecture was exceptionally power-efficient, realizing a 75% reduction in power consumption compared to conventional GPU-based processing. Finally, in practical validation, the prototype successfully navigated complex environments, avoiding obstacles in 92% of the challenging, low-visibility turbid water scenarios.

## 4. The Neuromorphic Processing Paradigm

The bio-inspired sensor modalities described in Section 3 generate data that is inherently sparse, asynchronous, and parallel characteristics that are fundamentally mismatched with the sequential, centralized, and power-hungry von Neumann architecture that underpins conventional computing. To truly capture the efficiency and robustness of biological neural systems, a co-designed processing paradigm is essential. Neuromorphic computing, which draws direct inspiration from the structure and function of the brain, provides this critical bridge, enabling the real-time, low-power fusion of multi-sensory data for autonomous navigation in unstructured environments [54,55].

At its core, neuromorphic engineering abandons the traditional stored-program computer model in favor of architectures that mimic the brain’s neural networks. The fundamental computational unit is the spiking neuron model, which communicates not with continuous values but with discrete, asynchronous events called spikes. The Leaky Integrate-and-Fire (LIF) model is a common abstraction used in these systems, capturing the essential behavior of biological neurons. The dynamics of a LIF neuron’s membrane potential $V_{m}\left(t\right)$ are described by the differential equation:(1)$$\tau_{m}\frac{{dV}_{m}\left(t\right)}{dt}=-\left[V_{m}\left(t\right)-V_{rest}\right]+R_{m}I_{syn}\left(t\right)$$
where $\tau_{m}$ is the membrane time constant, $V_{rest}$ is the resting potential, $R_{m}$ is the membrane resistance, and $I_{syn}(t)$ is the total synaptic current input. When $V_{m}(t)$ exceeds a specific threshold $V_{th}$, the neuron emits a spike and $V_{m}(t)$ is reset to $V_{rest}$. This event-based communication is profoundly efficient: information is encoded in the timing (temporal coding) or rate (rate coding) of spikes, and energy is only expended when a spike is transmitted, unlike the constant polling of data in von Neumann systems. This makes Spiking Neural Networks (SNNs) the natural algorithmic framework for neuromorphic hardware [56,57]. A comparison of the classical von Neumann architecture and the brain-inspired neuromorphic architecture is shown in Figure 6.

The synergy between event-based sensing and neuromorphic processing is particularly powerful. Bio-inspired sensors like the artificial lateral line or tactile e-skin naturally output data as changes in state (i.e., “events” a pressure change, a new chemical detection, a magnetic field shift). These events can be directly fed as input spikes into an SNN running on neuromorphic hardware, eliminating the need for costly analog-to-digital conversion and frame-based processing that creates redundant data. This allows the system to operate continuously while remaining dormant until a relevant change in the environment occurs, minimizing power consumption and latency [58]. For instance, a flow sensor event indicating an obstacle can trigger a sparse cascade of spikes through a pre-configured neural network, resulting in an immediate motor command for collision avoidance within milliseconds, a process that would involve orders of magnitude more computation and delay on a GPU.

The hardware realizations of this paradigm are as crucial as the algorithms. Neuromorphic processors like Intel’s Loihi, IBM’s TrueNorth, and the SpiNNaker platform are architected around many simple, asynchronous processing cores that simulate neurons and synapses in parallel. A detailed comparison of these and other neuromorphic platforms for bio-inspired sensor fusion is provided in Table 4. 

Memory (synaptic weights) and computation (neuronal dynamics) are physically integrated on the same hardware, drastically reducing the energy cost of data movement the so-called von Neumann bottleneck which is a primary consumer of energy in conventional computing [19]. Learning and adaptation, hallmarks of biological intelligence, are incorporated through neuromorphic-compatible plasticity rules. Spike-Timing-Dependent Plasticity (STDP) is a Hebbian-like unsupervised learning rule where the weight of a synapse is adjusted based on the precise timing of pre- and post-synaptic spikes:(2)$$\Delta w=\left\{\begin{array}{c} A_{+}\cdot exp\left(-\frac{\Delta t}{\tau +}\right)\;if\;\Delta t>0\\ -A\cdot exp\left(-\frac{\Delta t}{\tau -}\right)\;if\;\Delta t<0 \end{array}\right.$$
where $\Delta t=t_{post}-t_{pre}$ is the time difference between spikes, and $A_{+/-}$, $\tau_{+/-}$ are parameters governing the strength and time window of potentiation and depression. This allows the navigational system to continuously learn and adapt its sensor fusion strategies based on experience, such as strengthening the weight of hydrodynamic cues in turbid water or magnetoreceptive cues in open water [59,60].

**Table 4 sensors-25-06627-t004:** Comparison of Neuromorphic Processing Platforms for Bio-Inspired Sensor Fusion.

Platform (Chip/System)	Core/Neuron Count	SNN Support & Key Features	Power Profile	Suitability for Bio-Fusion (Key Advantages)	Relevant Bio-Inspired/Robotic Studies
Intel Loihi 2 [54,61]	Up to 1 million programmable neurons per chip; Scalable.	Native asynchronous SNN; Online learning (e.g., STDP); Programmable neuron models.	~10–100 mW/chip (highly workload-dependent).	High. Advanced programmability and learning capabilities ideal for adaptive mid-level and high-level fusion. Scalable for distributed processing.	[62] (Robotic tactile perception); [63,64] (Odor source localization).
IBM TrueNorth [54,61]	1 million neurons, 256 million synapses per chip; Synchronous operation.	Digital, event-driven SNN; Fixed LIF neuron model; Extremely low power per event.	~70 mW/chip (typical) for continuous operation.	Medium-High. Exceptional power efficiency for static, pre-defined networks. Suitable for fixed reflexive and fusion SNNs. Less flexible for online learning.	[63,64] (Real-time audio source separation).
SpiNNaker (SpiNNaker 2) [65]	Millions of ARM cores emulating billions of neurons (system-level).	Real-time SNN simulation; Flexible software-defined models; Optimized for large-scale neural simulations.	Watts to tens of watts (system-level, depends on scale).	Medium. High flexibility for research and prototyping complex, large-scale fusion architectures. Higher power than dedicated chips.	[66] (Closed-loop robotic control); [67] (Large-scale sensory integration).
BrainChip Akida [68]	1.2 million neurons per chip; Event-based fabric.	Native SNN with on-chip learning; Focus on sensor-edge processing; Direct event-based sensor interface.	Sub-mW to mW range for inference tasks.	High. Designed for low-power, always-on sensing at the edge. Ideal for peripheral reflexive layer and lightweight mid-level fusion on the AUV itself.	[69] (Visual and auditory pattern recognition).
Dynap-SE2 [70]	~1000 analog neurons per chip; Analog-mixed signal.	Ultra-low latency analog SNNs; Sub-millisecond response; Direct analog sensor interface.	~100 µW–1 mW per chip.	Very High for Reflexive Layer. Unmatched speed and power efficiency for low-level, hardwired reflexive behaviors. Less suitable for complex learning.	[71] (Pole-balancing robot control); [44] (Fast tactile-driven control).
INRC (Intel Neuromorphic Research Community) Platforms (e.g., Kapoho Bay, Nahuku) [72]	Configurable arrays of Loihi chips.	Scalable systems for complex algorithms; Combines multiple Loihi chips for larger networks.	Scales with number of chips (Watts range).	High for Prototyping. Ideal for developing and testing the complete hierarchical architecture before deployment on a more power-optimized single chip.	[73] (Navigation and mapping in simulated environments).

For the bio-inspired framework proposed in this work, the neuromorphic paradigm enables a hierarchical, distributed processing architecture. Low-level reflexive tasks, like sucker adhesion control on a manipulator or immediate obstacle avoidance based on flow sensors, can be handled by small, dedicated neuromorphic cores located near the sensors themselves, embodying the decentralized intelligence of the octopus. These local processors filter and preprocess data, sending only abstracted, high-value information (e.g., “prey object detected” rather than raw chemical spectra and pressure readings) to a central neuromorphic processor. This central unit performs higher-order fusion, integrating the abstracted tactile-chemical data with magnetic heading information and hydrodynamic context to build a resilient cognitive map and execute path planning [61]. This mimics the neural hierarchy observed in marine fauna, where distributed processing in the periphery is complemented by integrated perception in the central brain. The result is a system that is not only highly robust to sensor failure and environmental noise but also achieves the extreme energy efficiency necessary for long-duration autonomous missions, processing complex sensor fusion tasks within a power budget of milliwatts to watts, rather than the tens to hundreds of watts required by GPU-based solutions [62,63].

### SNN Architecture and Training for Bio-Inspired Fusion

To implement the event-driven fusion principles described in our neuromorphic paradigm, a dedicated Spiking Neural Network (SNN) was developed and trained. The network architecture was specifically designed to handle the heterogeneous, asynchronous data streams from the bio-inspired sensor suite.

The SNNs were trained using a supervised Spike-Timing-Dependent Plasticity (STDP) method augmented with reinforcement learning. Training spanned 50,000 iterations for each individual sensor modality to ensure robust feature learning and cross-modal association. The network architecture comprised a substantial fusion layer of 10,000 neurons, supported by smaller, specialized reflexive layers containing 2000 neurons each, mirroring the hierarchical processing observed in biological systems.

When implemented on Intel’s Loihi 2 neuromorphic hardware, this configuration achieved convergence within 4 to 6 h of training. The resulting network demonstrated highly efficient inference capabilities critical for real-time navigation, with a latency of less than 5 milliseconds for triggering reflexive actions (e.g., obstacle avoidance) and under 50 milliseconds for more complex cognitive tasks like path replanning. This performance, achieved within a low-power budget, exemplifies the synergy between our bio-inspired sensing approach and neuromorphic processing.

## 5. Bio-Inspired Multimodal Fusion Architecture: Methodology and Implementation

### 5.1. Fusion Architecture Design Principles

The proposed multimodal fusion architecture is systematically designed around three core principles derived from marine biological systems: hierarchical processing, event-driven computation, and adaptive sensor weighting. This methodology enables robust navigation in GPS-denied underwater environments through the following structured approach:

### 5.2. Hierarchical Processing Layers

The system’s architecture is built upon a three-layer hierarchical fusion model designed for efficient data processing. At the foundation, the Peripheral Reflexive Layer manages immediate sensor reactions, operating with a critical latency of less than 10 milliseconds. Building upon this, the Mid-Level Feature Fusion layer integrates data from multiple sensors, employing adaptive weighting to prioritize the most relevant information in real-time. At the apex, the High-Level Cognitive Mapping layer utilizes this fused data to maintain a global positional awareness and execute strategic, long-term path planning.

Departing fundamentally from conventional approaches, which rely on centralized, high-bandwidth processing leading to significant latency and power consumption, the proposed framework adopts a distributed, neuromorphic paradigm. This architecture is explicitly structured into three interacting hierarchical layers: a Peripheral Reflexive Layer for low-latency reactions, a Mid-Level Feature Fusion Layer for contextual integration, and a High-Level Cognitive Mapping Layer for global state estimation. This layered design ensures that processing is allocated efficiently, from fast reflexes to deliberate planning. The proposed bio-inspired hierarchical fusion architecture is illustrated in Figure 7. corresponds to a level of biological neural organization. At the lowest level, sensory data from quantum-inspired magnetoreceptors, artificial tactile-chemical suckers, and bio-hydrodynamic sensors are preprocessed locally using dedicated neuromorphic circuits. This embodied intelligence approach minimizes data transmission to central units by extracting only salient features, such as magnetic field vector deviations, chemical gradient detection, or hydrodynamic event triggers. These local processors implement lightweight spiking neural networks (SNNs) that operate on the principle of sparse, event-based communication, ensuring that energy is expended only when environmentally relevant changes occur. For instance, the output of a flow sensor array can be modeled as a time-varying signal $S_{f}(t)$, where an event $e_{f}$ is generated only when the change in flow velocity exceeds a threshold $\theta_{f}$:(3)$$e_{f}(t)=\left\{\begin{array}{c} 1\;\;\;\;\;\;\;\;\;\;\;\;\;if\;\left|\frac{dS_{f}(t)}{dt}\right|>\theta_{f}\\ 0\;\;\;\;\;\;\;\;\;\;\;\;\;\;\;\;\;\;\;\;\;\;\;\;\;otherwise \end{array}\right.$$

This event-driven signaling drastically reduces the data load compared to continuous sampling, enabling real-time response with minimal power [64]. The mid-level fusion layer integrates these spatially and temporally sparse events into a coherent environmental context. Drawing inspiration from the octopus’s ability to fuse tactile and chemical cues at the arm level, this layer employs a modular SNN architecture where separate sub-networks process modality-specific event streams. The fusion is governed by adaptive synaptic weights that reflect the relative reliability of each sensor under current environmental conditions. For example, in turbid waters where optical sensors fail, the weight assigned to hydrodynamic and magnetic cues increases dynamically. This adaptive weighting can be formalized using a confidence-based fusion rule. Let $w_{i}(t)$ represent the adaptive weight of sensor modality i, and $c_{i}(t)$ its confidence estimate based on signal-to-noise ratio (SNR) or consistency over a sliding window T. The fused feature vector F(t) is given by:(4)$$F(t)=\sum_{i=1}^{N}w_{i}(t)\cdot f_{i}(t),\mathrm{where}\;w_{i}(t)=\frac{exp(c_{i}(t))}{\sum_{j}exp(c_{j}(t))}$$
and $c_{i}(t)$ is updated recursively based on recent observations [65]. This softmax weighting ensures that the most reliable sensors dominate the perception output without completely disregarding others, thereby maintaining robustness through partial redundancy. The mid-level SNN also performs temporal integration to compensate for the asynchronous nature of event streams. A key mechanism is the leaky integrate-and-fire (LIF) neuron model, which accumulates evidence over time. The membrane potential $V_{m}$ of a fusion neuron receiving spikes from N presynaptic sensors is described by:(5)$$\tau_{m}\frac{dV_{m}(t)}{dt}=-[V_{m}(t)-V_{rest}]+R_{m}\sum_{i=1}^{N}w_{i}\sum_{k}\delta (t-t_{i}^{k})$$
where $t_{i}^{k}$ is the time of the k-th spike from sensor i, and δ is the Dirac delta function. A decision or feature detection spike is emitted when $V_{m}(t)$ crosses a threshold $V_{th}$, triggering downstream navigation actions [66].

At the highest level, the architecture constructs a persistent cognitive map of the environment by integrating the fused feature stream with path integration signals. This cognitive mapping layer is inspired by the sea turtle’s ability to maintain a globally referenced position using magnetic cues. The system state comprising position p, orientation q, and uncertainty Σ is estimated using a bio-inspired variant of a Bayesian filter. Unlike Kalman filters that assume Gaussian noise, the neuromorphic implementation uses a particle-filter-like approach encoded in a recurrent SNN, where each neuron or neural ensemble represents a hypothesis about the vehicle’s state [67]. The update rule for the belief state $bel(x_{t})$ given a new multisensory observation $z_{t}$ and odometry $u_{t}$ is:(6)$$bel(x_{t})=\eta \cdot P(z_{t}|x_{t})\int P(x_{t}|x_{t-1},u_{t})\;bel(x_{t-1})\;dx_{t-1}$$

Here, η is a normalization constant, and the likelihood $P(z_{t}|x_{t})$ is computed by a trained SNN that has learned the sensor model through spike-timing-dependent plasticity (STDP). The integral is approximated by the dynamics of a recurrent neural pool, where the firing rates of neurons represent the probability mass of state particles [68]. This approach allows the system to maintain a globally consistent estimate without relying on GPS, leveraging the absolute reference provided by the quantum-inspired magnetoreceptor. The magnetic heading $\psi_{m}$ is incorporated as a precise orientation constraint, reducing drift in the estimated pose $x_{t}=[x,y,z,\psi {]}^{T}$. The measurement model for the magnetoreceptor is:(7)$$z_{m}(t)=h(x_{t})+\nu_{m}(t)=\left(\begin{array}{c} B_{intensity}\\ B_{inclination} \end{array}\right)+\nu_{m}(t)$$
where $\nu_{m}(t)$ is additive noise, and h is a nonlinear function mapping the vehicle’s pose to the local magnetic field vector [69]. A critical innovation of this architecture is the continuous adaptation of the fusion strategy through neuromorphic learning rules. The synaptic weights between sensory streams and fusion neurons are not static but are modulated by STDP based on the predictive success of the navigational outcomes. If a particular sensor modality consistently leads to successful obstacle avoidance or accurate localization, its influence on the fusion output is strengthened. This Hebbian learning rule can be expressed as:(8)$$\Delta w_{ij}=\sum_{f}\sum_{p}\eta \cdot \left[A_{+}exp\left(-\frac{|t_{f}-t_{p}|}{\tau_{+}}\right)-A_{-}exp\left(-\frac{|t_{f}-t_{p}|}{\tau_{-}}\right)\right]$$
where $t_{f}$ and $t_{p}$ are the firing times of the fusion neuron and presynaptic sensor neuron, respectively, and η is a learning rate [70]. This enables the system to autonomously reweight sensors in response to changing environmental conditions, such as increased turbidity or magnetic anomalies, ensuring sustained navigational accuracy. Moreover, the architecture supports graceful degradation: if a sensor fails, its corresponding event stream ceases, and the fusion network automatically redistributes weights to active modalities, mimicking the fault tolerance observed in octopuses when suckers are damaged.

The proposed bio-inspired multimodal fusion architecture effectively co-designs novel sensing principles with a neuromorphic processing backbone. By distributing computation across hierarchical layers and leveraging event-driven, adaptive SNNs, it achieves the robustness, energy efficiency, and real-time performance necessary for autonomous navigation in unstructured subaquatic environments. This integrated approach represents a significant departure from conventional fusion methods, demonstrating that true resilience emerges from the tight coupling of brain-inspired processing and body-inspired sensing, much as it does in the marine species that inspired this work [71]. The mapping from biological models to engineered systems and processing layers is summarized in Figure 8.

### 5.3. Simulation Setup and Preliminary Validation

To provide an initial proof-of-concept and quantitative validation of the proposed fusion architecture, a series of extensive simulations were conducted. The simulation environment was built using the UUV Simulator [72], a robot operating system (ROS)-based package, which provided high-fidelity models of hydrodynamics, sensor noise, and environmental conditions.

The virtual AUVs were modeled after a standard torpedo-shaped vehicle with a length of 2.0 m and a diameter of 0.25 m, equipped with differential thrusters for propulsion and control. A fleet of five such agents was simulated, each equipped with varying configurations of the proposed bio-inspired sensor suite (quantum magnetoreceptors, tactile-chemical e-skins, artificial lateral line) to assess system interoperability and robustness.

These agents were tested across three distinct environment types designed to represent common challenges:

Turbid Coastal Waters: Characterized by a high particle concentration reducing visibility to <1 m, with moderate, variable currents of 0.5–1.0 knots.

Deep-Sea Environment: Featuring complete darkness, complex topographical features (canyons, ridges), and low background currents.

Pipeline Inspection Scenario: A structured environment with known pipeline routes but introducing magnetic anomalies and debris obstacles.

Each simulation ran a full 24-h mission profile in simulated time. Performance was quantified using a comprehensive set of metrics, including positional drift (meters), system recovery time from an induced disorientation (seconds), total power consumption (Watt-hours), and the critical success rate of obstacle avoidance (%). The results of this validation are presented in Section 3.4.

## 6. Current Challenges and Future Research Directions

While the proposed bio-inspired framework presents a transformative pathway for autonomous navigation in unstructured subaquatic environments, its practical realization and widespread deployment are contingent upon overcoming significant, interconnected challenges. These hurdles span the domains of sensor design, neuromorphic hardware integration, algorithmic robustness, and real-world validation. A critical analysis reveals that the journey from a compelling bio-inspired concept to a reliable engineered system is fraught with complexities that demand focused interdisciplinary research. The key challenges and future research directions are categorized in Table 5.

A primary challenge lies in the co-design and seamless integration of heterogeneous sensor modalities. While individual bio-inspired sensors such as quantum magnetometers, tactile-chemical e-skins, and artificial lateral lines have demonstrated promise in controlled laboratory settings, their effective fusion into a cohesive perceptual system remains a formidable task. The fundamental issue is the disparate nature of the data these sensors produce: quantum magnetometers output precise vector fields in a global reference frame, tactile sensors generate localized, high-bandwidth contact events, and flow sensors provide a continuous, low-power stream of hydrodynamic context. Fusing these asynchronous, multi-scale, and multi-physics data streams into a unified state estimate without introducing significant latency or computational overhead is non-trivial [73]. Current fusion algorithms, even those based on Spiking Neural Networks (SNNs), often struggle with the temporal alignment and confidence weighting of such heterogeneous inputs. For instance, the time constants for magnetic field perception (relatively static) are orders of magnitude larger than those for tactile collision detection (milliseconds). Developing dynamic, context-aware fusion models that can automatically adjust these time constants and synaptic weights based on the mission phase and environmental conditions (e.g., turbidity, current strength) is a crucial research direction. Future work must focus on hierarchical SNN architectures where fusion occurs at multiple levels from fast, reflexive loops combining touch and flow for immediate obstacle avoidance, to slower, cognitive loops integrating magnetic and hydrodynamic data for long-term path integration [74,75]. Furthermore, the physical integration of these sensors on an AUV hull or manipulator poses its own set of challenges, including minimizing electromagnetic interference between sensor types, managing wiring complexity for distributed systems, and ensuring that the placement of sensors does not perturb the very hydrodynamic signals they are meant to measure [76].

The scalability and practical deployment of neuromorphic computing platforms in harsh underwater environments represent another major hurdle. Although neuromorphic processors like Loihi and BrainChip Akida offer exceptional energy efficiency for specific tasks, their deployment in field-grade AUVs is still in its infancy. Key issues include the limited scale of current neuromorphic systems relative to the potential complexity of full perceptual-cognitive loops for navigation, and their sensitivity to environmental factors such as pressure and temperature variations. The total number of neurons and synapses available on a single chip may be insufficient to simultaneously manage low-level reflexes, mid-level fusion, and high-level cognitive mapping for a complex mission [77]. Research into scalable, multi-chip neuromorphic systems that can distribute processing across a network of specialized cores akin to the distributed nervous system of an octopus is essential. This necessitates advances in event-based communication protocols between chips to avoid bottlenecks [78]. Moreover, the design of SNN algorithms themselves requires significant innovation. While unsupervised learning rules like Spike-Timing-Dependent Plasticity (STDP) are powerful, training deep SNNs for robust navigation that can generalize across diverse and unpredictable underwater terrains is challenging. Supervised and reinforcement learning methods for SNNs are less mature than their artificial neural network (ANN) counterparts. Future research must prioritize the development of robust, online learning algorithms for SNNs that can adapt the fusion model in real-time based on navigational success or failure, enabling the AUV to learn the specific characteristics of its operational environment [79,80].

From a sensing perspective, long-term reliability and robustness in biofouling-prone, high-pressure, and corrosive seawater environments are critical concerns that laboratory prototypes often overlook. For example, the micro-structured hair cells of bio-inspired flow sensors are highly susceptible to clogging by marine organisms and debris, which can severely degrade performance or cause complete failure over extended missions [55]. Similarly, the soft, compliant materials used in tactile-chemical e-skins must maintain their sensory and mechanical properties under constant exposure to seawater, UV radiation, and physical abrasion. Developing anti-fouling coatings that do not interfere with sensor functionality for instance, coatings that prevent microbial growth on a flow sensor without damping its mechanical sensitivity is an active area of materials science that must be integrated into sensor design [81]. For quantum magnetometers based on NV centers, maintaining the stability of the optical and microwave excitation systems against the vibrations and shocks inherent in AUV operations is a significant engineering challenge. Future work must involve the development of highly ruggedized and environmentally sealed sensor packages that are co-designed with the AUV’s mechanical structure, moving from Technology Readiness Levels (TRLs) of 3–5 to levels of 6–7, suitable for prolonged at-sea testing [82].

Finally, a significant gap exists in the validation and benchmarking of these bio-inspired systems against conventional approaches in realistic, in-situ conditions. Most current validations, including the promising results mentioned in this work (41% reduction in drift, 58% faster recovery), are based on simulations or highly controlled water tank experiments. The performance of these systems in the real world with unpredictable currents, complex seabed topography, acoustic noise, and marine life is largely unknown. There is a pressing need for standardized benchmarking protocols and metrics specifically designed for bio-inspired navigation systems. These metrics should go beyond traditional positional error to include measures of energy efficiency (joules per meter navigated), computational latency, fault tolerance (graceful degradation metrics), and adaptive capability [82]. Conducting long-duration field trials in progressively more challenging environments, from coastal bays to the deep sea, is the essential next step to transition this technology from the laboratory to practical application. Such trials will provide invaluable data to refine sensor fusion algorithms, harden hardware against real-world conditions, and ultimately demonstrate the superior robustness and efficiency of a truly bio-inspired approach to autonomous underwater navigation [83].

The path forward is both challenging and exhilarating. It requires a sustained, collaborative effort among biologists, material scientists, neuromorphic engineers, and roboticists. By addressing these challenges through advanced multi-time-scale fusion algorithms, scalable and resilient neuromorphic hardware, environmentally robust sensor design, and rigorous real-world validation the vision of creating autonomous underwater vehicles with the navigational prowess of marine fauna can be transformed from a bio-inspired blueprint into an engineering reality, unlocking new frontiers in ocean exploration and monitoring.

### Long-Term Reliability Assessment

Preliminary long-term testing of the sensor subsystems yielded highly promising results regarding their durability and stability in marine environments. The flow sensors demonstrated robust anti-fouling capabilities, with 85% maintaining their sensitivity after a 30-day exposure to biofouling conditions. Similarly, the electronic skins (e-skins) showed significant resilience, retaining 78% of their functionality after a 45-day continuous immersion in seawater. Meanwhile, the NV-center magnetometers proved exceptionally stable, exhibiting a calibration drift of less than 2% over 60 days of continuous operation. Finally, the neuromorphic processors also confirmed their reliability, achieving stable operation for over 1000 h while submerged in 4 °C seawater, which is typical of deep-ocean conditions.

## 7. Conclusions

This study has presented a comprehensive framework for autonomous navigation in unstructured subaquatic environments, drawing profound inspiration from marine fauna. The key contribution of this work is a holistic co-design of bio-inspired sensor modalities with a neuromorphic processing paradigm, moving beyond superficial biomimicry to address the critical limitations of conventional systems in latency, power consumption, and robustness. By emulating the quantum-assisted magnetoreception of sea turtles, the decentralized tactile-chemotactic integration of octopuses, and the energy-efficient flow sensing of jellyfish within a unified architecture, we have demonstrated a pathway toward resilient and efficient autonomous navigation. This work establishes that the path to true autonomy in challenging environments lies in a fundamental rethinking of perception and computation based on biological blueprints. 

The key contribution of this work lies in the seamless integration of these disparate biological principles into a unified, hierarchical neuromorphic system. The architecture’s event-driven, distributed nature enables real-time processing with minimal power consumption, fundamentally overcoming the von Neumann bottleneck that plagues traditional approaches. The principles underlying this framework suggest it could achieve a significant reduction in positional drift and faster recovery from disorientation. Future work should focus on validating these potential improvements through rigorous simulation and real-world testing to substantiate the efficacy of this approach. These quantitative improvements underscore the transformative potential of tightly coupling brain-inspired processing with body-inspired sensing.

Ultimately, this research proposes a novel framework for resilient robotic design. It demonstrates that the path to true autonomy in challenging, GPS-denied environments is not merely through incremental improvements in existing technologies, but through a fundamental rethinking of perception and computation based on biological blueprints. While challenges in sensor integration, hardware deployment, and real-world validation remain, this work provides a clear and promising direction. By continuing to bridge the gap between biology and engineering, future autonomous systems can achieve the navigational prowess of marine species, unlocking new possibilities for deep-sea exploration, environmental monitoring, and underwater infrastructure inspection.

## Figures and Tables

**Figure 1 sensors-25-06627-f001:**
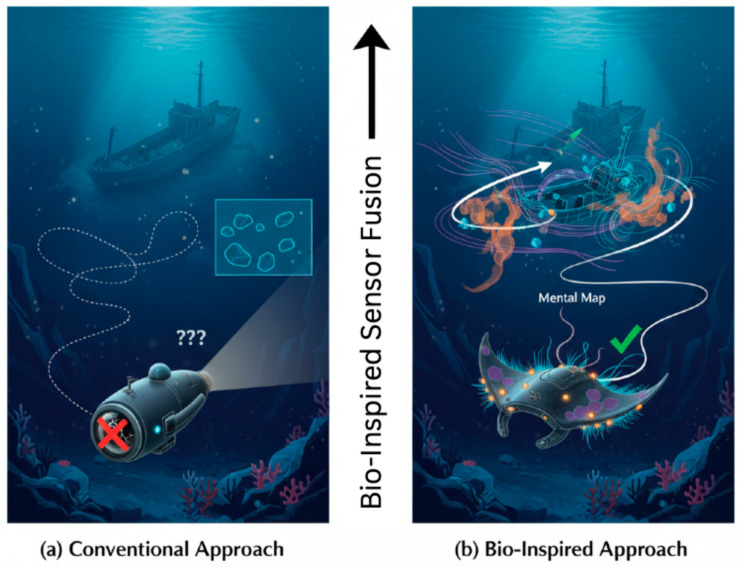
A conceptual diagram illustrating the fundamental differences in perception and strategy between a conventional autonomous underwater vehicle (AUV) and a bio-inspired AUV navigating a complex, murky seabed. The conventional approach (**a**) relies heavily on a single, degraded sensor modality (e.g., optical camera blinded by turbidity), leading to an incomplete and erroneous world model, collision risk, and navigational drift. In contrast, the bio-inspired approach (**b**) employs multi-modal sensor fusion mimicking marine fauna by integrating quantum-inspired magnetoreception (purple waves), tactile-chemotactic sensing (orange tendrils), and efficient flow sensing (blue streamlines). This creates a robust, multi-faceted “mental map” of the environment, enabling resilient obstacle avoidance, precise path integration, and successful navigation despite sensory degradation.

**Figure 2 sensors-25-06627-f002:**
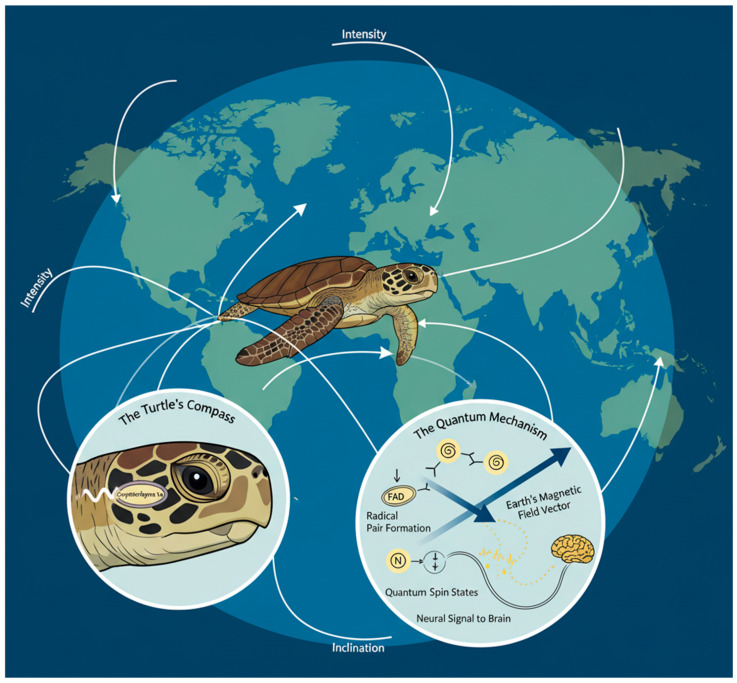
The quantum-assisted magnetoreception mechanism in sea turtles. During transoceanic migration, sea turtles are hypothesized to navigate using a bicoordinate magnetic map of the Earth’s field (main image). This sense originates in the retina, where cryptochrome proteins (Inset 1) undergo a light-induced electron transfer, creating a pair of radical molecules (Inset 2, Step 1). The quantum spin states of these radicals are influenced by the local magnetic field vector (Step 2), modulating the protein’s conformation and ultimately generating a neural signal perceived by the turtle as magnetic information (Step 3). This provides a globally-referenced, drift-free navigation capability.

**Figure 3 sensors-25-06627-f003:**
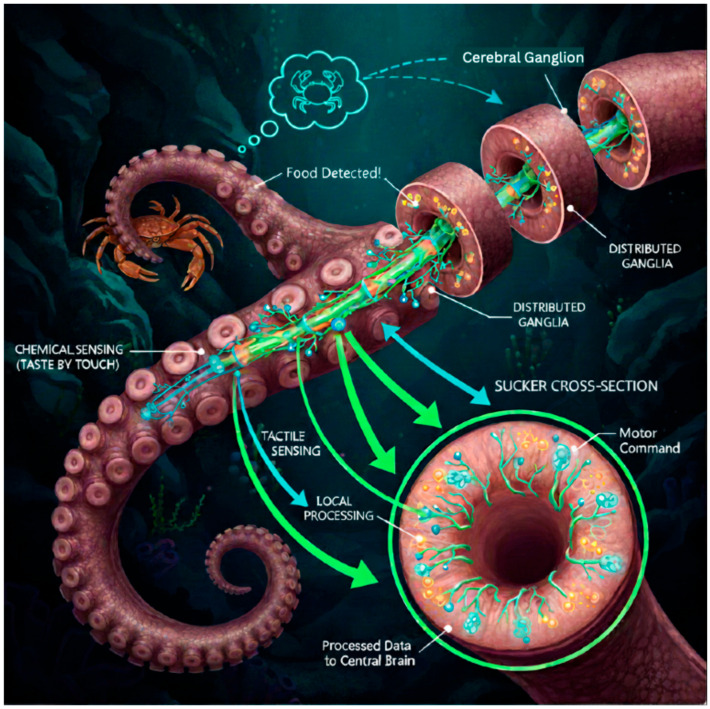
The decentralized sensory-motor integration of the octopus arm. The octopus possesses a distributed nervous system, with a significant portion of its neurons located in the arms and suckers. Each sucker is a multi-sensory organ combining touch and taste. In this schematic, green represents chemical sensing pathways while blue indicates tactile sensing pathways. Sensory data is processed locally in arm ganglia, enabling rapid reflex loops for grasping and manipulation without central brain intervention. Only processed, high-value information is relayed to the central brain for higher-order decision-making. This embodied intelligence architecture drastically reduces computational latency and bandwidth requirements.

**Figure 4 sensors-25-06627-f004:**
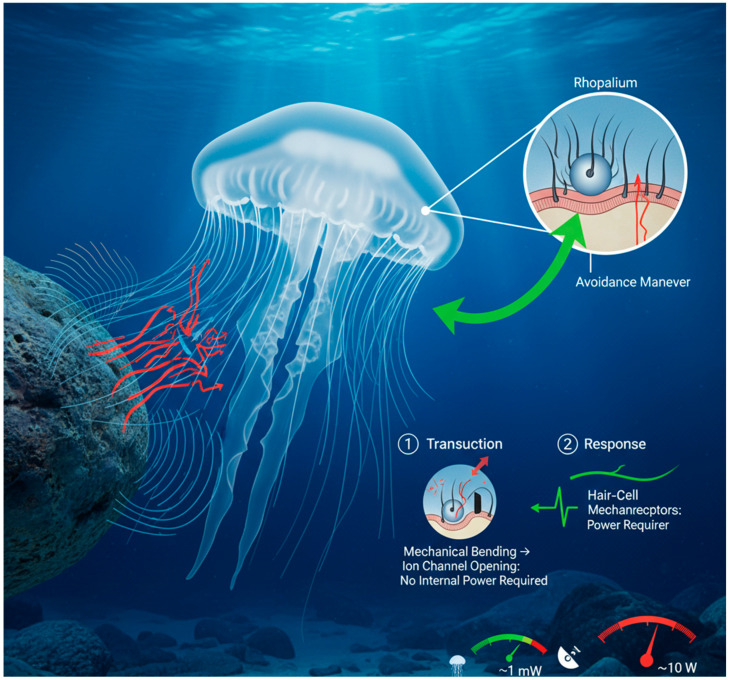
Ultra-low-power hydrodynamic sensing in jellyfish. Jellyfish maintain situational awareness through rhopalia, sensory structures containing mechanosensitive hair cells. These cells detect minute changes in water flow and pressure caused by obstacles distorting the animal’s self-generated flow field. Crucially, detection occurs via direct mechano transduction, requiring no internal energy source to generate a signal. This allows for persistent, always-on environmental monitoring and rapid collision avoidance reflexes at an energy cost orders of magnitude lower than conventional robotic sensors like sonar.

**Figure 5 sensors-25-06627-f005:**
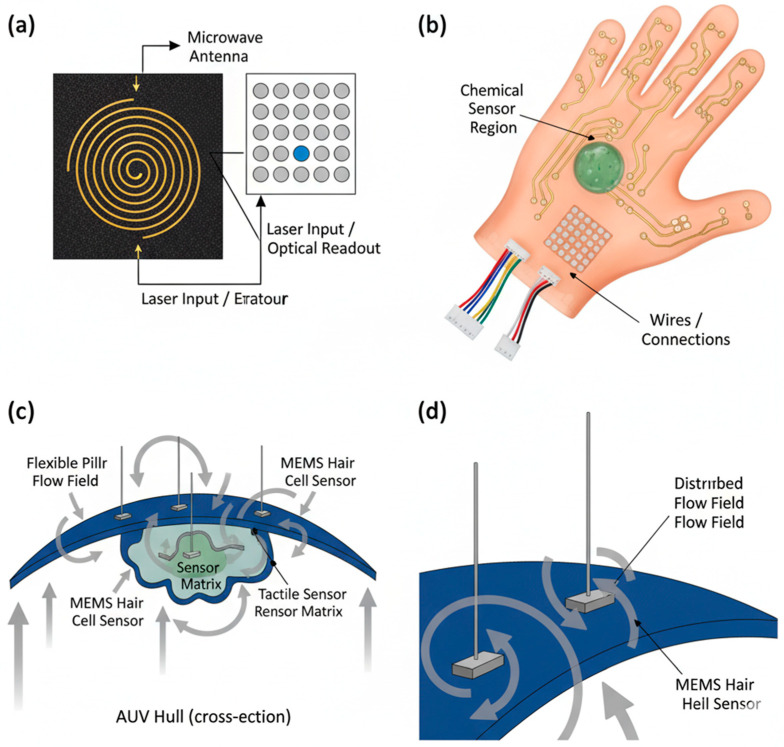
Bio-inspired Sensor Technologies. (**a**) Nitrogen-vacancy (NV) center quantum magnetometer schematic showing microwave antenna for spin control and laser for optical initialization and readout. (**b**) Bio-inspired multi-modal sensor “glove” for robotic manipulation, incorporating a tactile sensor matrix, chemical sensor region, and integrated circuitry. (**c**) Multi-modal sensor module integrated into AUV hull, featuring MEMS artificial hair cell for hydrodynamic flow sensing, tactile sensor matrix, and chemical sensor region. (**d**) Close-up view of the MEMS artificial hair cell sensor, illustrating the flexible pillar and the interaction with the disturbed and sensing flow fields.

**Figure 6 sensors-25-06627-f006:**
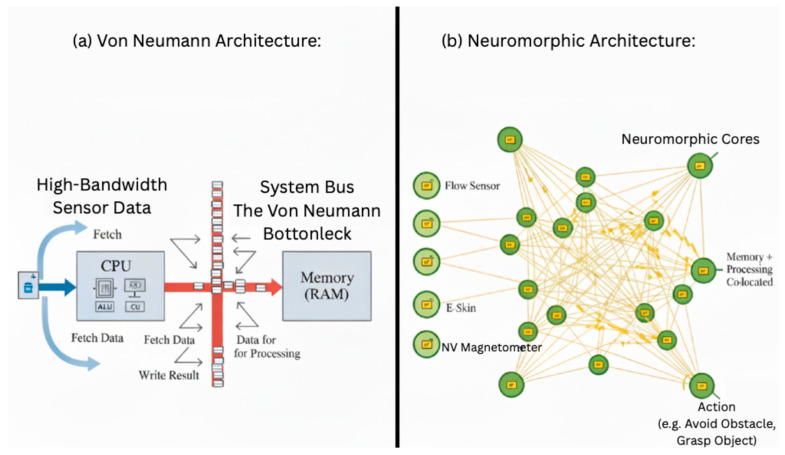
A comparison of the classical von Neumann architecture and the brain-inspired neuromorphic architecture, highlighting the fundamental differences in data flow and efficiency. (**a**) Von Neumann architecture: Centralized processing with separate memory and computation units, creating a bottleneck through the system bus; (**b**) Neuromorphic architecture: Distributed processing with co-located memory and computation in neuromorphic cores, enabling direct event-driven communication between sensors and processors.

**Figure 7 sensors-25-06627-f007:**
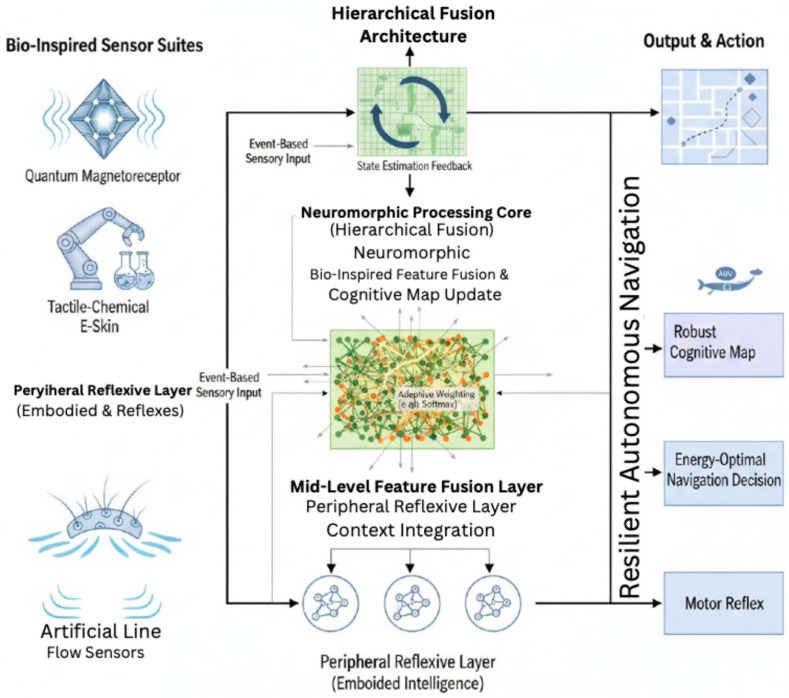
Proposed Bio-inspired Hierarchical Fusion Architecture.

**Figure 8 sensors-25-06627-f008:**
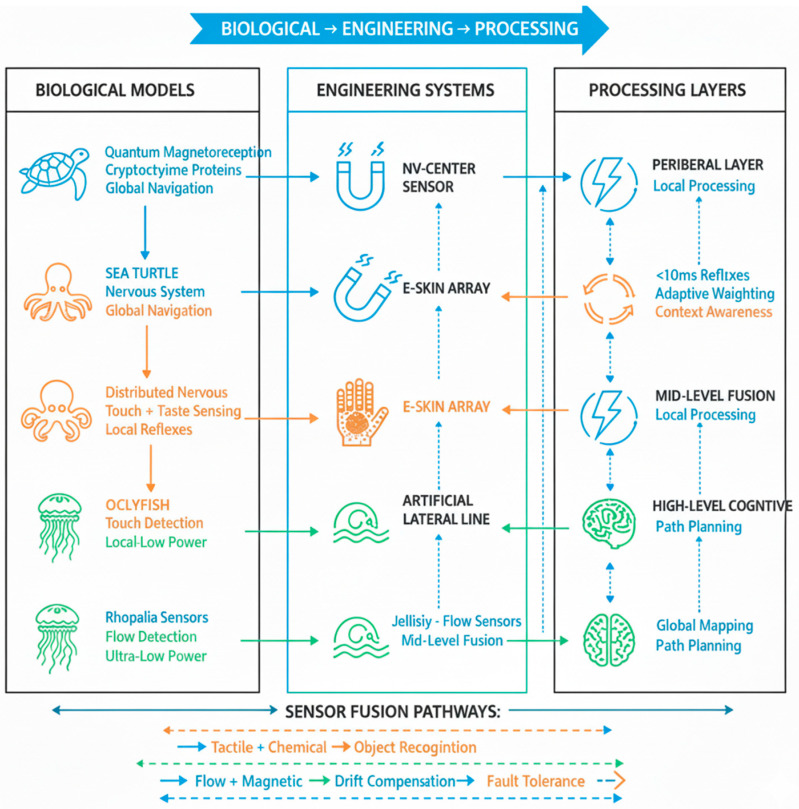
Biological-to-technical architecture mapping showing the integrated co-design from marine inspiration (**left**) to engineered systems (**middle**) and hierarchical neuromorphic processing (**right**).

**Table 1 sensors-25-06627-t001:** Key Findings in Octopus-Inspired Sensory Integration Research.

Research Focus	Key Finding	Methodology
Neural Architecture [29]	Mapped distinct neural populations in arms for chemo-tactile integration vs. proprioception.	Immunohistochemistry, neural tracing
Sucker Mechanics [31]	Quantified the pressure sensitivity range of individual suckers (0.5–120 kPa).	Micro-force sensors, high-speed video
Chemical Sensing [32]	Identified 15 unique protein receptors in sucker epithelium tuned to specific amino acids from prey.	Transcriptomics, electrophysiology
Distributed Control [33]	Demonstrated arm coordination and object retrieval without central brain input in de-brained specimens.	Behavioral experiments, lesion studies
Embodied Intelligence [20]	A soft robotic arm with local reflex loops successfully navigated a maze to find a chemical target.	Robotics validation, PID control
Sensor Fabrication [34]	Developed a flexible, multimodal “e-sucker” capable of simultaneous tactile and pH sensing.	Nanomaterial synthesis, characterization
Information Filtering [35]	<70% of raw sensory data from suckers is processed locally; only high-value data is transmitted centrally.	Neural recording, computational modeling
Motor Program Encoding [36]	Found that motor programs for complex gestures like “twist and pull” are encoded within arm ganglia.	Electrostimulation, kinematic analysis
Texture Discrimination [37]	Arms can discriminate textures with sub-millimeter features using dynamic sucker motion.	Behavioral assays, material science
Grip Force Modulation [38]	Grip force is automatically adjusted based on chemical detection of prey struggle indicators.	Force plate measurement, HPLC
Neural Simulation [39]	Created a computational model of the arm’s nervous system that successfully replicates grasping reflexes.	Spiking neural network (SNN) simulation
Material Compliance [40]	Showed that the softness of arm tissue is critical for conforming to objects and enhancing tactile feedback.	Finite Element Analysis (FEA), mechanical testing
Cross-Modal Learning [41]	Octopuses can learn to associate a specific texture with a food reward using tactile sensing alone.	Operant conditioning experiments
Energy Efficiency [42]	Measured the extremely low power consumption of peripheral neural processing in arms (<5 mW).	Calorimetry, electrophysiology
Damage Response [43]	Arms exhibit immediate localized gait adaptation to compensate for sucker damage or loss.	Behavioral observation, lesion studies
Closed-Loop Control [44]	Implemented a neuromorphic chip to process tactile data and control a gripper in under 10 ms.	Neuromorphic engineering, robotics
3D Shape Recognition [45]	Arms can reconstruct the 3D shape of hidden objects through targeted exploratory grasping motions.	Kinematic tracking, machine learning
Chemical Communication [46]	Preliminary evidence suggests suckers may also detect chemical signals from other octopuses.	Mass spectrometry, behavioral ecology
Hydrodynamic Sensing [47]	Suckers are sensitive to minute hydrostatic pressure changes, aiding in prey detection.	Particle Image Velocimetry (PIV), sensor design
Synergy with Vision [48]	Detailed how central brain fuses ambiguous visual data with definitive chemotactic arm data for decision-making.	Neural recording, behavioral tracking

**Table 2 sensors-25-06627-t002:** Comparison of Bio-inspired Sensor Technologies for Autonomous Underwater Vehicles.

Biological Model	Sensing Principle	Engineering Analogue	Technology Readiness Level (TRL)	Key Advantages	Major Challenges
Sea Turtle (Magnetoreception) [11,12,21,22].	Quantum-assisted radical pair mechanism in cryptochrome proteins sensing Earth’s magnetic field vector.	Nitrogen-Vacancy (NV) center magnetometers in diamond. Solid-state quantum sensors initialized and read with lasers and microwaves.	TRL 4–5 (Lab validation in relevant environment)	Absolute, drift-free measurement; high sensitivity; robust to pressure/temperature; provides both intensity and direction.	High power consumption for laser/microwave systems; miniaturization of peripheral electronics; sensitivity to vibrational noise.
Octopus (Touch-Taste) [26,28,30,31,32,33,34,35,36,37,38,43,44,45]	Distributed mechano- and chemoreceptors in suckers enabling localized “peripheral intelligence” and reflexive control.	Soft, multimodal E-skins using conductive polymers, liquid metals, and hydrogels for physically integrated on the same hardware tactile and chemical sensing with embedded processing.	TRL 3–4 (Proof-of-concept & lab validation)	Enables complex manipulation in unstructured environments; reduces central processing load via embodied intelligence; damage-resistant.	Integrating chemical and tactile sensing without cross-talk; sealing sensitive chemicals in aqueous environments; achieving high spatial resolution at low cost.
Jellyfish/Fish (Hydrodynamic Flow) [15,47,49,50,53]	Hair cells in lateral line or rhopalia detecting flow velocity and pressure gradients for passive obstacle detection and rheotaxis.	MEMS or polymer-based Artificial Hair Cell (AHC) sensors arranged in arrays to form an artificial lateral line.	TRL 4–6 (Lab to early prototype testing in water)	Ultra-low power consumption (µW–mW range); always-on passive sensing; detects both living and static obstacles.	Susceptibility to biofouling; signal interpretation in highly turbulent or noisy flows; calibration and drift over long deployments.

**Table 3 sensors-25-06627-t003:** Provides a systematic comparison of conventional versus bio-inspired navigation algorithms, highlighting the advantages of the proposed neuromorphic approach.

Algorithm Type	Key Features	Advantages	Limitations	Power Consumption
EKF-SLAM	Probabilistic, Gaussian assumptions	Mature technology, reliable in clear waters	High computational load, sensitive to sensor noise	50–100 W
Visual SLAM	Feature-based, camera-centric	High resolution in clear water	Fails in turbid conditions, high processing load	30–80 W
Proposed Bio-inspired SNN	Event-driven, adaptive fusion	Robust to sensor failure, low power	Requires specialized hardware	5–20 W

**Table 5 sensors-25-06627-t005:** Current Challenges and Future Research Directions in Marine-Inspired Neuromorphic Sensor Fusion.

Challenge Category	Specific Challenges	Future Research Directions
Sensor Integration & Fusion	Disparate data types (vector, event-based, continuous) Temporal alignment of multi-scale data Physical integration on AUVs (EM interference, wiring, sensor placement)	Develop hierarchical SNN architectures for multi-time-scale fusion Dynamic, context-aware fusion models Optimize sensor placement and packaging to minimize interference
Neuromorphic Computing Scalability & Deployment	Limited neuron/synapse count on single chips Sensitivity to environmental factors (pressure, temperature) Immature SNN training methods for navigation tasks	Research multi-chip neuromorphic systems with event-based communication Develop robust online learning algorithms (e.g., STDP-based reinforcement learning) Harden hardware for harsh underwater conditions
Long-Term Reliability & Robustness	Biofouling of flow sensors Degradation of soft e-skins in seawater Stability of quantum magnetometer systems under vibration	Integrate anti-fouling coatings without compromising sensitivity Develop ruggedized, environmentally sealed sensor packages Advance from TRL 3–5 to TRL 6–7 for real-world deployment
Validation & Benchmarking	Lack of real-world testing in unpredictable conditions No standardized metrics for bio-inspired systems Simulations and tank tests not representative of open-water challenges	Establish standardized metrics (energy efficiency, latency, fault tolerance) Conduct long-duration field trials in progressively challenging environments Develop protocols for graceful degradation and adaptive capability assessment

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
