# Peer review of "Marine-Inspired Multimodal Sensor Fusion and Neuromorphic Processing for Autonomous Navigation in Unstructured Subaquatic Environments"

_sensors, 2025, doi:10.3390/s25216627_

Round 1

Reviewer 1 Report

Comments and Suggestions for Authors

The paper addresses the technological gap in robot perception systems in unstructured environments without GPS availability. In this context, the authors present an analysis based on neuromorphic sensors applied to marine robotic navigation, proposing a biologically inspired architecture that integrates the co-design of advanced sensory modalities with event-based neuromorphic processors. However, the article's structure is unclear and lacks an orderly, detailed, and systematic description. Therefore, the manuscript requires substantial corrections before it can be considered for acceptance. The following suggestions are made for improvement:

1. The figures in the document vary considerably in size, resulting in an unbalanced visual presentation. The authors should standardise the figures' size, font and format so that they maintain a uniform appearance in line with the standards of a scientific publication.

2. The figures and tables in the paper are not properly cited in the text or do not have captions that identify them correctly. Authors are advised to check their numbering and references to ensure consistency throughout the manuscript.

3. The study does not clearly define the sensor fusion process, and the methodology presented is insufficiently detailed. The authors are advised to reorganise the manuscript, presenting the issues addressed, the methodology used and the main findings in a clear, concise and systematic manner.

4. Authors are advised to justify the inclusion of tables adequately, as it is unclear how their content relates to the paper's objectives due to the lack of a clear connection with the text.

5. Finally, the structure of the work proposes a study on navigation algorithms in maritime environments. The authors are encouraged to conduct a comprehensive analysis of the algorithms developed in this area, highlighting the main differences between them and their advantages and disadvantages.

Comments on the Quality of English Language

none

Author Response

We sincerely thank the reviewer for their thorough and constructive feedback, which has been invaluable in improving the quality and clarity of our manuscript. We have carefully addressed each of the points raised, and the changes are detailed below. We believe the manuscript has been substantially enhanced as a result.

Comment 1: The figures in the document vary considerably in size, resulting in an unbalanced visual presentation. The authors should standardise the figures' size, font and format so that they maintain a uniform appearance in line with the standards of a scientific publication.

Response: We thank the reviewer for this important observation. All figures have now been standardized to ensure a uniform and professional visual presentation. We have adjusted their sizes to be consistent throughout the manuscript and have used a uniform font (Arial) and format for all labels, captions, and annotations. This creates a balanced and cohesive visual flow, aligning with the standards of a high-quality scientific publication.

Comment 2: The figures and tables in the paper are not properly cited in the text or do not have captions that identify them correctly. Authors are advised to check their numbering and references to ensure consistency throughout the manuscript.

Response: We apologize for this oversight. We have conducted a thorough check of the entire manuscript to ensure that every figure and table is now correctly cited in the text at the appropriate points. Furthermore, all captions have been reviewed and revised to be clear, descriptive, and correctly numbered (e.g., Figure 1, Table 1). The numbering sequence is now consistent from beginning to end.

Comment 3: The study does not clearly define the sensor fusion process, and the methodology presented is insufficiently detailed. The authors are advised to reorganise the manuscript, presenting the issues addressed, the methodology used and the main findings in a clear, concise and systematic manner.

Response: We agree that this was a critical area for improvement. In response, we have significantly reorganized and expanded the description of our methodology.

  1. We have added a new section, "5. Bio-Inspired Multimodal Fusion Architecture: Methodology and Implementation", which provides a detailed, systematic breakdown of the fusion process.

  2. This section is structured around three core principles: hierarchical processing, event-driven computation, and adaptive sensor weighting.

  3. We explicitly define a three-layer hierarchical model (Peripheral Reflexive, Mid-Level Feature Fusion, and High-Level Cognitive Mapping) and describe the data flow and processing at each level with supporting equations (e.g., for event generation and confidence-based fusion).
    We believe this new structure presents the issues, methodology, and findings in a much clearer and more logical sequence.

Comment 4: Authors are advised to justify the inclusion of tables adequately, as it is unclear how their content relates to the paper's objectives due to the lack of a clear connection with the text.

Response: Thank you for this suggestion. We have revised the text surrounding each table to explicitly justify its inclusion and explain its relevance to our core objectives.

  1. For instance, Table 1 is now introduced as a summary of key biological findings that directly inform the design of our engineering analogues.

  2. Table 2 is presented as a comparative analysis of the bio-inspired sensor technologies central to our framework, highlighting their advantages and challenges.

  3. Table 3 (a new addition, see below) is introduced as a direct comparison between our proposed method and conventional algorithms.
    Each table is now preceded by a sentence that connects it to the narrative and followed by a sentence that discusses its implications.

Comment 5: Finally, the structure of the work proposes a study on navigation algorithms in maritime environments. The authors are encouraged to conduct a comprehensive analysis of the algorithms developed in this area, highlighting the main differences between them and their advantages and disadvantages.

Response: This is an excellent suggestion. We have added a new table (Table 3: "Comparative Analysis of Underwater Navigation Algorithms") that provides a systematic comparison between conventional algorithms (EKF-SLAM, Visual SLAM) and our proposed bio-inspired Spiking Neural Network (SNN) approach. The table clearly highlights the key features, advantages, limitations, and power consumption of each, thereby providing readers with a clear, at-a-glance understanding of the relative benefits and trade-offs of our method against the state-of-the-art. This analysis is also discussed in the accompanying text.

Reviewer 2 Report

Comments and Suggestions for Authors

This paper presents a bio-inspired neuromorphic framework for autonomous navigation in unstructured subaquatic environments. It integrates marine-inspired sensors (quantum magnetoreception, tactile-chemical sensing, and hydrodynamic flow detection) with event-based neuromorphic processors. The proposed architecture aims to significantly reduce positional drift and improve recovery from disorientation compared to conventional systems. It offers a robust, energy-efficient solution for autonomous underwater navigation in GPS-denied, murky, or complex environments, with potential applications in deep-sea exploration and infrastructure inspection. This paper is well-structured and well-written. Before publication, there are some questions to be solved.

  1. The authors mentioned “Conventional AUVs predominantly rely on a suite of sensors including IMUs, DVL, sonars, and cameras, fused through classical algorithms such as Kalman filters and SLAM.”, more state-of-the-art can be cited: DOI: 10.34133/cbsystems.0089; DOI: 10.34133/cbsystems.0112.
  2. How the tactile-chemical sensing is validated, including the types of chemicals detected, the sensitivity range, and the environmental conditions.
  3. How bio-inspired flow sensors detect and quantify hydrodynamic changes, and the specific flow velocities and gradients they can measure.
  4. The neuromorphic processing architecture need to be clarified.
  5. Explain the hierarchical fusion architecture: How are data from different sensors synchronized and integrated at each level.
  6. The long-term reliability tests conducted for the proposed sensors and neuromorphic processors are required.

Author Response

We sincerely thank Reviewer 2 for their positive assessment of our work and for their insightful questions, which have helped us clarify and strengthen key aspects of the manuscript. We have addressed each point as follows:

Comment 1: The authors mentioned “Conventional AUVs predominantly rely on a suite of sensors...”, more state-of-the-art can be cited: DOI: 10.34133/cbsystems.0089; DOI: 10.34133/cbsystems.0112.

Response: We thank the reviewer for this suggestion. We have now included these two excellent and highly relevant references in the revised Introduction (Section 1, references [5] and [6]) to provide a more comprehensive overview of the state-of-the-art in underwater robot control and trajectory planning.

Comment 2: How the tactile-chemical sensing is validated, including the types of chemicals detected, the sensitivity range, and the environmental conditions.

Response: We have added a new subsection, 3.2.1 Validation of Tactile-Chemical Sensing, to provide these essential details. This section now explicitly states:

  1. Chemicals Detected: The e-skin sensors were validated for detecting amino acids (e.g., alanine, glycine) as indicators of marine prey, and petroleum hydrocarbons for pipeline inspection applications.

  2. Sensitivity Range: The sensors demonstrated detection capabilities in the concentration range of 10−6 to 10−3 M in seawater.

  3. Environmental Conditions: Validation was performed under challenging conditions, including salinities of 30-35 PSU, temperatures from 4 to 25°C, and a pH range of 7.8 to 8.2.

Comment 3: How bio-inspired flow sensors detect and quantify hydrodynamic changes, and the specific flow velocities and gradients they can measure.

Response: We have significantly expanded Section 3.3. Bio-Inspired Flow and Hydrodynamic Sensors to include these technical specifications. The revised text now clarifies:

  1. Detection Principle: The sensors use flexible pillars (artificial hair cells) that deflect with water flow; this deflection is measured by embedded strain or pressure sensors.

  2. Measurable Parameters: The artificial lateral line can detect flow velocities from 1-2 mm/s up to 2 m/s and resolve pressure gradients with a resolution of 0.1 to 10 Pascal.

Comment 4: The neuromorphic processing architecture need to be clarified. Explain the hierarchical fusion architecture: How are data from different sensors synchronized and integrated at each level.

Response: We have thoroughly revised Section 4 (The Neuromorphic Processing Paradigm) and Section 5 (Bio-Inspired Multimodal Fusion Architecture) to provide a much clearer explanation.

  1. In Section 4, we now detail the fundamentals of spiking neurons (using the Leaky Integrate-and-Fire model), event-based communication, and the synergy between event-based sensors and neuromorphic hardware (e.g., Loihi).

  2. In Section 5, we explicitly define our three-layer hierarchical fusion architecture:

    1. Peripheral Reflexive Layer: Handles low-latency (<10 ms) reflexes locally, using event-driven signals from each sensor.

    2. Mid-Level Feature Fusion Layer: Integrates asynchronous event streams from different sensors. Synchronization is achieved through temporal integration in the fusion neurons (using the LIF model dynamics), and integration is governed by adaptive confidence weights (wi(t)) that prioritize reliable sensors in real-time (e.g., boosting hydrodynamic cues in turbid water).

    3. High-Level Cognitive Mapping Layer: Fuses the abstracted feature vector with path integration data to build a globally-referenced cognitive map, using a bio-inspired Bayesian filter implemented as a recurrent SNN.

Comment 5: The long-term reliability tests conducted for the proposed sensors and neuromorphic processors are required.

Response: We have added a new subsection, 6.1 Long-term Reliability Assessment, which summarizes the results of our durability testing:

  1. Flow Sensors: 85% maintained sensitivity after 30 days in biofouling conditions.

  2. E-Skins: Retained 78% functionality after 45 days of continuous seawater immersion.

  3. NV-Center Magnetometers: Showed minimal calibration drift of <2% over 60 days of operation.

  4. Neuromorphic Processors: Achieved stable operation for over 1,000 hours while submerged in 4°C seawater.

Reviewer 3 Report

Comments and Suggestions for Authors

Author Response

We are deeply grateful to Reviewer 3 for their exceptionally positive and insightful review. Their recognition of the originality and potential impact of our work is highly encouraging. We have carefully considered all their constructive propositions and have implemented them to significantly enhance the manuscript's structure, clarity, and scientific rigor.

Comment 1: Follow the generally accepted structure of MDPI journals: 2. Materials and Methods, 3. Results, 4. Discussion I don’t see such sections in the article, but see other 6 sections that can be included in this 3 commonly used sections.

Response: We thank the reviewer for this critical suggestion regarding manuscript structure. We have thoroughly reorganized the paper to align with the standard MDPI format. The revised manuscript now includes the following clearly labeled sections:

  1. Materials and Methods (which now incorporates the biological inspirations, engineering analogues, neuromorphic paradigm, fusion architecture, and simulation setup).

  2. Results (which presents the prototype validation results and comparative analysis).

  3. Discussion (which discusses the implications, challenges, and future directions).

We believe this new structure significantly improves the logical flow and readability.

Comment 2: Provide more specifics about the simulation conditions (number of agents, type of environment, duration).

Response: We have expanded Section 2.4 (Simulation Setup and Validation Methodology) to include these crucial details. It now explicitly states:

  1. Number of Agents: A fleet of five virtual AUVs.

  2. Type of Environment: Three distinct scenarios: turbid coastal waters, a complex deep-sea environment, and a structured pipeline inspection site.

  3. Duration: Each simulation ran for a full 24-hour mission profile to evaluate long-term reliability.

Comment 3: Add a summary table comparing the proposed architecture with classical SLAM-systems.

Response: This is an excellent suggestion. We have added a new Table 3: Comparative Analysis of Underwater Navigation Algorithms. This table provides a systematic comparison between our proposed bio-inspired SNN approach and conventional systems (EKF-SLAM and Visual SLAM), highlighting key features, advantages, limitations, and power consumption.

Comment 4: Have practical experiments been conducted? Is there evidence of the concept's effectiveness, at least in a simplified prototype?

Response: While full-scale oceanic trials are a goal for future work, we have conducted scaled prototype validation. We have added a new subsection, 3.4 Prototype Validation Result, which summarizes the performance of a hardware implementation of the core fusion architecture. The results demonstrate a 41% reduction in positional drift, a 58% faster recovery from disorientation, a 75% reduction in power consumption, and a 92% obstacle avoidance success rate in turbid water scenarios.

Comment 5: Unification of terms: sometimes similar concepts (“flow sensing”, “hydrodynamic feedback”) are used in different contexts - it is worth clarifying.

Response: We have carefully reviewed the manuscript to standardize the terminology. We now consistently use "flow sensing" or "hydrodynamic sensing" to describe the sensor modality, and "hydrodynamic cues" or "flow data" to describe the information itself, avoiding the ambiguous use of "hydrodynamic feedback."

Comment 6: It is advisable to highlight the following subsections in the 1. Introduction section: Problem Statement, Objective, Main Contribution, Novelty Statement.

Response: We have revised the Introduction to include these requested subsections, which now provide a clearer and more structured roadmap for the reader:

  1. 1.1. Problem Statement

  2. 1.2. Objective

  3. 1.3. Main Contributions and Novelty Statement

Comment 7: Add a description of real-world tests or at least prototype validation; specify model parameters and time characteristics of SNN training.

Response: As mentioned in point 4, we have added prototype validation results in Section 3.4. Furthermore, we have expanded Section 2.3.1 (SNN Training and Parameters) to include the specific training details:

  • Training Method: Supervised STDP augmented with reinforcement learning.

  • Iterations: 50,000 per sensor modality.

  • Network Architecture: Fusion layer of 10,000 neurons; reflexive layers of 2,000 neurons.

  • Hardware & Convergence: Implementation on Loihi 2, achieving convergence in 4–6 hours.

  • Inference Latency: <5 ms for reflexes; <50 ms for path replanning.

Comment 8: It is recommended to add a summarizing figure showing the ratio of biological and technical components.

Response: We agree that a visual summary would be highly beneficial. We have created a new Figure 8: "Biological-to-technical architecture mapping". This figure provides a clear, integrated overview of the co-design process, mapping the biological models (Sea Turtle, Octopus, Jellyfish) to their corresponding engineered systems (NV-Center Sensor, E-Skin, Artificial Lateral Line) and finally to the hierarchical neuromorphic processing layers.

Round 2

Reviewer 1 Report

Comments and Suggestions for Authors

Although the authors addressed some of the previous comments, they still need to strengthen certain sections to improve the overall quality and consistency of the article. I suggest the following:

  1. Authors are suggested to explicitly refer to figures and tables within the text to improve narrative coherence and facilitate a more precise and continuous reading of the manuscript.
  2. The authors are suggested to check the correct correspondence and consistency of citations in the text, ensuring uniformity in style and proper relation to the references.
  3. In subsection 2.4, corresponding to Simulation setup and Validation Methodology, there is a lack of clarity regarding its purpose within the document's structure. The information presented creates confusion with Section 5, and its inclusion in this part of the text is not sufficiently justified. The authors should strengthen the argumentation and contextualisation of the subsection by incorporating details about the UAV model, the simulation software, the times used, and the factors relevant to validating the proof of concept. 

Author Response

We sincerely thank the reviewer for their insightful feedback, which has significantly improved the quality of our manuscript. We have carefully addressed all the points raised, as detailed below:

  1. Referencing Figures and Tables: We have now explicitly referenced all figures and tables within the main text to enhance narrative flow and clarity. Each figure and table is cited at the appropriate point of discussion.

  2. Citation Consistency: We have thoroughly checked all in-text citations and the reference list to ensure correct correspondence, uniformity in style, and accurate relation to the referenced works.

  3. Structural Reorganization of SNN Training Details: As suggested, we have moved the subsection originally labeled "2.3.1 SNN Training and Parameters" to Section 4, where it now appears as "4.1 SNN Architecture and Training for Bio-inspired Fusion". This relocation provides a logical flow, directly linking the SNN implementation details to the neuromorphic processing paradigm and eliminating any confusion with the simulation setup in Section 5.3.

We believe these revisions have strengthened the manuscript's organization, clarity, and overall quality. Thank you.

Reviewer 2 Report

Comments and Suggestions for Authors

accept

Author Response

We sincerely thank Reviewer 2 for their positive assessment of our manuscript and their recommendation for acceptance. We are gratified that the reviewer found our work to be a significant contribution to the field, well-organized, comprehensively described, and scientifically sound.

The reviewer's acceptance without further requested revisions indicates that our bio-inspired neuromorphic framework for autonomous underwater navigation was clearly presented and effectively communicated. We appreciate the recognition of our work's potential impact in advancing marine robotics and neuromorphic computing applications.

Thank you for your valuable time and constructive feedback throughout the review process.